# Moistube irrigation (MTI) discharge under variable evaporative demand

**Tinashe Lindel Dirwai**[1,2]*, **Aidan Senzanje**[1], **Tafadzwanashe Mabhaudhi**[3,4]

**1** School of Engineering, University of KwaZulu-Natal, Pietermaritzburg, KwaZulu-Natal, South Africa,
**2** VarMac Consulting Engineers, Pietermaritzburg, KwaZulu-Natal, South Africa, **3** Center for Transformative Agricultural and Food Systems, School of Agricultural, Earth and Environmental Sciences, University of KwaZulu-Natal, Pietermaritzburg, KwaZulu-Natal, South Africa, **4** Centre for Water Resources Research, School of Agricultural, Earth and Environmental Sciences, University of KwaZulu-Natal, Pietermaritzburg, KwaZulu-Natal, South Africa

* tldirwai@gmail.com

**Data Availability Statement:** Methods and protocols for MTI discharge under pressure discharge under variable evaporative demand are available as a collection in protocols.io (https://dx.doi.org/10.17504/protocols.io.bmdrk256).

## Abstract

We investigated the conceptual capability of Moistube irrigation (MTI) to discharge under zero applied positive pressure and under varied climatic conditions by inducing an artificial evaporative demand ($E_d$) or negative pressure around Moistube tubing. This study was premised on the null hypothesis that an artificially induced $E_d$ or negative pressure does not impact MTI discharge. Moistube tubing was enclosed in a 1 m long PVC conduit. A 20 l water reservoir placed on an electronic balance provided a continuous supply of water whilst a three-speed hot air blower facilitated the radiative factor and advection process. The procedure was conducted under varied climatic conditions with three air velocity ($u_a$) treatments namely; 1.2 m.s$^{-1}$, 2.5 m.s$^{-1}$, and 3.0 m.s$^{-1}$ and the experiment run times were 159 h, 134 h and 10 h, respectively. The average temperature ($T_{ave}$) and relative humidity (RH) data for $u_a$ = 1.2 m.s$^{-1}$ were 53°C and 7.31%, whilst for $u_a$ = 2.5 m.s$^{-1}$, $T_{ave}$ was 56°C and RH = 7.19%, and for $u_a$ = 3.0 m.s$^{-1}$, $T_{ave}$ was 63°C and RH = 6.16%. The experimental data was input into the four variable Penman-Monteith method to compute the evaporative demand ($E_d$). For each $E_d$, the instantaneous mass flow rate ($\dot{m}$) was recorded using an electronic balance and subsequently converted to volumetric flow rates. For each of the air velocities, the respective $E_d$ values obtained were 0.16, 0.31 and 0.36 mm.d$^{-1}$. The Bowen ratios ($r$) were well below 1 ($r < 1$), which suggested a sufficient supply of moisture to evaporate. For $E_d$ = 0.16 mm.d$^{-1}$ the vapour pressure deficit (VPD) was 113.08 mbars, whilst for $E_d$ = 0.31 mm.d$^{-1}$ and for $E_d$ = 0.36 mm.d$^{-1}$ the VPD were 129.93 mbars and 150.14 mbars, respectively. The recorded discharges ($q$) at normalised time ($t^*$) = 1 h for $E_d$ = 0.16 mm.d$^{-1}$ was 7.67*10$^{-3}$ l.hr$^{-1}$.m$^{-1}$ length, whilst for $E_d$ = 0.31 mm.d$^{-1}$ $q$ = 14.5*10$^{-3}$ l.hr$^{-1}$.m$^{-1}$ length, and for $E_d$ = 0.36 mm.d$^{-1}$ $q$ = 20.8*10$^{-3}$ l.hr$^{-1}$.m$^{-1}$ length. The imposed negative pressure causes an exponential increase in Moistube™ discharge, thus disproving the null hypothesis. The higher the evaporative demand the higher the discharge. This phenomenon allows MTI to be used for deficit irrigation purposes and allows irrigators to capitalize on realistic soil matric potential irrigation scheduling approach.

**Funding:** This work was supported by the Water Research Commission of South Africa (WRC -SA) – Drainage Project and by the University of KwaZulu-Natal – Pietermaritzburg, South Africa. The funder provided support in the form of a research stipend for author TLD and access to laboratory equipment, and VarMac Consulting Engineers provided support in the form of salary for author TLD. The funders did not have any additional role in the study design, data collection and analysis, decision to publish, or preparation of the manuscript. The specific roles of the authors are articulated in the 'author contributions' section.

**Competing interests:** The authors have read the journal's policy and have the following potential competing interests: TLD is a paid employee of VarMac Consulting Engineers. This does not alter our adherence to PLOS ONE policies on sharing data and materials. There are no patents, products in development or marketed products associated with this research to declare. Furthermore, we unequivocally state that the commercial entity did not play a role in the study design, data collection and analysis, decision to publish, or preparation of the manuscript.

# 1 Introduction

Moistube irrigation (MTI) is a relatively new semi-permeable membrane (SPM) irrigation technology. A typical third generation Moistube pipe has an outer protective membrane and an inner membrane that constitutes of densely and uniformly spaced nano-pores whose pore-diameter ranges from 10–900 nm. The technology utilises nano-technology such that the inner membrane imitates plant water uptake, which facilitates discharge according to crop water requirements [1, 2]. MTI is a low pressure discharge sub-surface irrigation technology whose functionality is similar to ceramic pitcher pots. Under a negative pressure, or in the absence of applied pressure, the discharge is a function of matric potential ($\psi$) [1–3].Negative pressure irrigation (NPI) is when water supply pressure is regulated by soil matric potential [4]. NPI can be classified as a precision irrigation technique which offers benefits such as continuous regulation of soil moisture thus improving crop yield, reduction of non-beneficial water use such as water loss by evaporation and runoff [4–6]. Conceptually, when water potential ($\psi_{water}$) is greater than the matric potential of the surrounding soil ($\psi_{soil}$), the MTI discharge rate is high and vice versa [1, 3]. There exists a number of issues in need of research answers, for example, how MTI discharge varies with imposed evaporative demand ($E_d$). The $E_d$ mimics changing soil water conditions thus exploring MTI applicability to deficit irrigation.

According to FAO [7], evapotranspiration ($ET$) is a combination of water loss from soils and transpiration by plants; the water loss mechanisms occur simultaneously under ambient conditions. A simpler method of estimating $ET$ is by using evaporative demand ($E_d$), which is defined as the upper boundary for $ET$ under ambient conditions with an uncapped hydrological limit or unlimited water supply. $E_d$ is used in irrigation scheduling as a proxy for plant water consumptive use wherein crop coefficients or factors such as phenology and soil stress are used to estimate the $ET_o$ [8]. There exists a correlation between rate and amount of plant water use and $E_d$ [9], thus since MTI discharge is a function of soil matric potential a high $E_d$ potentially increases MTI discharge. Currently there is a dearth in literature concerning MTI discharge under variable $E_d$. What is known is how rate and amount of water uptake correlates with $E_d$ [8].

$E_d$ has three drivers, which are the hydrological, radiative and the advective limits [8, 10]. For evaporative demand to occur there has to be an adequate water supply to meet the minimum hydrological limit, hence, the hydrological limit defines the availability of water to evaporate and transpire from plants and soil surfaces. According to Hobbins and Huntington [8], the radiative limit is the energy required to facilitate the evaporative process Under field conditions, the radiative limit is influenced by variables such as surface albedo ($\alpha$), and shortwave and longwave radiation. For a buried MTI lateral the soil heat fluxes are also involved. Advection is a critical component for $E_d$. The advective limit describes the system boundary's ability to absorb and bear away moisture.

Limited research efforts have been made to model $E_d$ under controlled and varied micro-climatic conditions. For example, Donohue, McVicar [10] used five evapotranspiration formulations namely Penman, Priestley–Taylor, Morton point, Morton areal and Thornthwaite to model and assess the best proxy for $E_d$. Abu-Zreig, Zraiqat [11] modelled an artificially induced pan evaporation scenario in order to measure the discharge rates of ceramic pitchers under negative head. The theoretical discharge design aspects of the MTI technology have not been tested. Understanding negative pressure discharge capability of MTI tubing can potentially aid irrigators to inform deficit irrigation strategies and save on energy costs that otherwise drive positive head irrigation systems, and more importantly assess MTI ability to satisfy the irrigation requirements of high water demand crops such as sugar cane.

The study investigated the conceptual discharge mechanism of Moistube™ irrigation when subjected to a negative pressure or in the absence of a positive driving pressure. The study hypothesized that the presence of an imposed negative pressure or an artificial $E_d$ cannot induce MTI discharge. This study adds to knowledge by providing answers around the conceptual zero pressure head discharge capability of MTI. The $E_d$ was used to simulate low matric potential or negative soil water tension conditions whilst monitoring the subsequent discharge performance of a buried Moistube tubing.

## 2 Materials and methods

Methods and protocols for MTI discharge under pressure discharge under variable evaporative demand are available as a collection in protocol.io (https://dx.doi.org/10.17504/protocols.io.bmdrk256).

### 2.1 Study site

The experiment was carried out at the University of KwaZulu-Natal Hydrology Laboratory (29.626044, 30.403325). The laboratory had a controlled room temperature of 22°C ± 1°C and a measured relative humidity (RH) of 55% ± 5%. Controlled conditions helped to eliminate the variations in atmospheric temperature and humidity.

### 2.2 Experimental design and set-up

The experiment was a single factor experiment: air velocity ($u_a$) with three controlled air velocities namely 1.2 m.s$^{-1}$, 2.5 m.s$^{-1}$ and 3.0 m.s$^{-1}$. Each air velocity level was replicated three times and it was dictated by the default hot air blower settings. The experiment comprised five recorded variables, namely, relative humidity (RH), air velocity ($u_a$), net radiation ($R_n$), water mass flow rate ($\dot{m}$), and micro-climate temperature ($T_a$). For each $u_a$, the subsequent $T_a$ and $R_n$, were recorded at five-minute intervals for 159 hr, 134 hr and 10 hr, respectively. The last experimental run was limited to 10 hr because the experienced temperatures exceeded realistic temperature scenarios that a buried MTI lateral can experience. The VPD expressed as the difference between actual vapour pressure of the air ($e_a$) and the observed vapour pressure ($e$) was derived from the recorded $T_a$ and $R_n$. Each replicated $u_a$ yielded values that were used to compute the resultant average $E_d$ and the corresponding average $\dot{m}$ values. The $\dot{m}$ was converted to discharge ($q$) by multiplying the recorded values by the density of water ($\rho_w$).

The equipment was assembled as shown in Fig 1. Air was blown axially to the suspended 1 m long Moistube™ lateral tubing in the PVC conduit. The flow of hot air was provided by a three speed 1800 W hot air blower. The water mass flow ($\dot{m}$) was measured using a GFK 75H electronic balance with a resolution of 0.001 kg (Adam Equipment, South Africa). The water level in the reservoir was kept constant and at the same elevation as the MTI lateral to eliminate the effect of water pressure head on discharge. The relative humidity (RH) was measured using the HCT01-00D sensor (E + E ELEKTRONIC ™, Austria) with a 5–95% RH working range, resolution of ± 2.5% RH, 2% RH accuracy, and a temperature dependency of ± 0.03% RH/°C, meaning the measured values were ± 0.02 close to the actual value and a limit of detection of ± 0.025. The temperature was measured using thermocouples (J-type) and a Pt1000 sensor (E + E ELEKTRONIC ™, Austria) with a resolution of ± 0.3°C and an accuracy of 0.1°C. Air velocity was measured using a hot film anemometer (EE 65 Series, Austria) with a working range of 0 m.s$^{-1}$–20 m.s$^{-1}$ and a resolution and accuracy of ± 0.2 m.s$^{-1}$. The sensors were connected to a five terminal unit and 12 channel VGR-B100 (RKC Instrument ™, Japan) data logger. The logger was programmed to record average data every five-minute interval for each experimental $u_a$ and the subsequent replications.

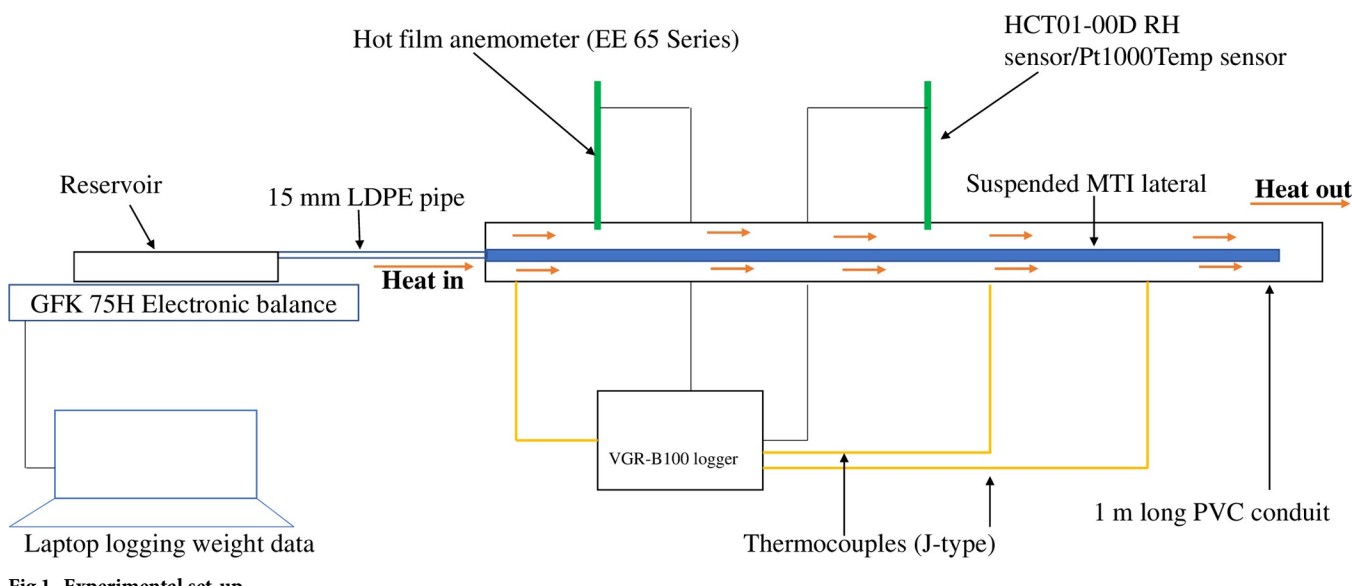

**Fig 1. Experimental set-up.**

## 2.3 ET model selection

The following *ET* models were assessed as proxies for determining $E_d$: (i) Morton areal, (ii) Morton point, (iii) Penman-Monteith, (iv) Priestley–Taylor, and (v) Thornthwaite (Table 1). These models are considered as universal standards for estimating *ET* [12]. Model selection was based on the ability to accommodate all measured variables. The Penman-Monteith model proved satisfactory since the model could accommodate the input data variables obtained from the experimental data i.e., $T_a$, $(e_a−e)$, $R_n$, and $u_a$.

**2.3.1 Penman-Monteith model application to compute $E_d$.** The standardized ASCE Penman-Monteith model was used to compute $E_d$. Standardized formulation aided in retaining data accuracy yet simplifying applicability. The model is defined by Eq 1 [7, 14, 17, 18]:

$$E_d = \frac{0.408\Delta(R_n - G) + \gamma \frac{C_n}{T+273} u_a(e_a - e)}{\Delta + \gamma(1 + C_d u_a)}$$

where: $\Delta$ = slope of the saturation vapour-pressure curve at air temperature (kPa.°C), $R_n$ = net radiation (W.m⁻²), $e_a−e$ = vapour-pressure deficit (mbars), $e$ = actual vapour pressure of air (mbars), $\gamma$ = psychometric constant of proportionality (kPa.°C⁻¹), $T$ = hourly air temperature (°C), $C_n$ = numerator constant for reference type and calculation time step ($C_n$ = 900), and $C_d$ = denominator constant for reference type and calculation time ($C_d$ = 0.34) [17].

**Table 1. $E_d$ formulations and the respective variables applied in the model(s).**

| Formulation | Variables | Reference |
|---|---|---|
| Morton areal | $T_a$, $(e_a−e)$,, $R_n$ | [10, 13] |
| Morton point | $T_a$, $(e_a−e)$,, $R_n$ | [10, 13] |
| Penman-Monteith | $T_a$, $(e_a−e)$,, $R_n$, $u_a$ | Penman [14] cited by Donohue, McVicar [10] |
| Priestley–Taylor | $T_a$, $R_n$ | Priestley and Taylor [15] cited by Donohue, McVicar [10] |
| Thornthwaite | $T_a$ | Thornthwaite [16] cited by [10] |

The energy balance equation was used to determine the evaporation energy. The $R_n$ was determined by Eq 2 and it was recorded in five-minute intervals:

$$R_n = R_i - R_r - L_u - R_c - R_s + R_a \qquad\qquad 2$$

where: $R_i$ = short wave solar radiation, $R_r$ = reflected part of the solar energy, $R_c$ is the conduction energy in air, $R_s$ = incremental stored energy in the conduit, $L_u$ = longwave radiation flux, and $R_a$ = advective energy. $R_i$, $R_c$, $R_s$ and $R_a$ are measured in flux units (W.m$^{-2}$). Parameters $R_i$, $R_r$, $L_u$, and $R_c$ were neglected because the experiment was conducted in the laboratory. According to Cross [17] when $R_s$ and $R_a$ are measured in five-minute intervals which constitute "short intervals" the two parameters can be neglected since they are negligible. Furthermore, considering that the experiment was not a closed air system, it therefore invalidated the relevance of $R_s$. In addition, the parameter $R_s$ is dependent on duration of sunshine hours (DS) and maximum possible sunshine hours available whilst the parameter $R_a$ utilises the $d_r$ function which is the distance between the earth and the sun [19], thus providing further evidence in their exclusion in Eq 2. Therefore $R_n$ was computed as per Eq 3:

$$R_n = \frac{Q_{in}}{\rho_w * L * (1 + r)} \qquad\qquad 3$$

where $Q_{in}$ = energy from the heater (J) (Eq 4), $\rho_w$ = density of water (kg.m$^{-3}$), $L$ = latent heat of evaporation (J.kg$^{-1}$), and $r$ = Bowen ratio (Eq 5):

$$Q_{in} = \rho_{air}(\dot{v}A) * C_p(\Delta T) + \varepsilon\sigma\Delta T^4 + hA\Delta T \qquad\qquad 4$$

where $A$ = area of the blower duct (m$^2$), $\dot{v}$ = air volumetric flow rate (m$^3$.s$^{-1}$) (product of the cross sectional area ($A$) of hot air blower duct and the average flow velocity $u_i$), $C_p$ = specific heat capacity of air (J.Kg$^{-1}$.$^\circ$C$^{-1}$), $\varepsilon$ is the emissivity coefficient of the PVC conduit (0.92) [20], $\sigma$ = Stefan-Boltzmann Constant (5.670367$^*$10$^{-8}$ W.m$^{-2}$$^\circ$C$^{-4}$), $h$ = convective heat transfer coefficient (W.m$^{-2}$$^\circ$C$^{-1}$), $\rho_{air}$ = density of air (kg.m$^{-3}$), and $\Delta T$ = change in temperature along the PVC conduit heating surface ($^\circ$C). The Bowen ratio was determined as follows:

$$r = \frac{6.1 * 10^{-4} * \rho_{air} * \Delta T}{e_a - e} \qquad\qquad 5$$

The $q$ vs $t$ relationship was established on a normalised time-scale ($t^*$). The $t^*$ was calculated according to Eq 6.

$$t^* = \frac{t - t_{min}}{t_{max} - t_{min}} \qquad\qquad 6$$

where: $t^*$ = normalised timed, $t$ = time variable to be normalised, $t_{max}$ and $t_{min}$ = maximum and minimum experimental run times to be normalised.

## 2.4 Statistical analyses

A normality test was undertaken on the three discharge data sets ($q_t$) obtained from the respective $E_d$ experiments using the Shapiro-Wilk normality test followed by a non-parametric Kruskal-Wallis one-way ANOVA test. All statistical analyses were done using R Studio© [21]. A linear regression of observed and simulated values was plotted and tested for goodness of fit using the $R^2$ value. To evaluate the functional relationship's performance the study employed the PBIAS statistics (Eq 7). The PBIAS statistic measured the degree of over or under-

**Table 2. Air velocity ($u_a$) measurements and the corresponding evaporative demand ($E_d$).**

| Air velocity ($u_a$) (m.s$^{-1}$) | Bowen ratio ($r$) | VPD ($e_a$−$e$) (mbars) | $E_d$ (mm.d$^{-1}$) |
|---|---|---|---|
| 1.2 | $3.502*10^{-6}$ | 113.081 | 0.16 |
| 2.5 | $3.221*10^{-6}$ | 129.934 | 0.31 |
| 3.0 | $3.135*10^{-6}$ | 150.144 | 0.36 |

estimation by the simulation model.

$$PBIAS = \frac{\sum_{i=1}^{x}(O_i - P_i) * 100}{\sum_{i=1}^{x} O_i}$$

Where $O_i$ and $P_i$ = observed and predicted value(s), respectively, $\bar{O}_i$ = mean observed data, and $x$ = number of observations.

## 3 Results and discussion

### 3.1 Air velocity ($u_a$) and evaporative demand ($E_d$)

The $E_d$ values for each air velocity are presented in Table 2. The Bowen ratios ($r$) were significantly low, which means the system had a sufficient water supply. This concurs with Hobbins and Huntington [8] and Cross [22] who posited that a Bowen ratio ($r$) of less than one signifies an unlimiting hydrological supply i.e., the MTI tubing, which is a non-water surface was relatively wet and had ample moisture to evaporate.

The results (Table 2) revealed a positive correlation between the variable $u_a$, and the other variables $r$, VPD, and $E_d$. A high air velocity ($u_a$) effected a high VPD, which subsequently induced a high $E_d$. The observation concurs with Donohue, McVicar [10] whose study attributed high evaporation rates to increased air temperatures. The study also established a negative correlation between $r$ and $E_d$ which signified that under a low $u_a$ evaporation occurred under a relatively and continuously wet surface i.e., wet environment evaporation ($E_w$). A high $u_a$ increased the drying power of air thus limiting readily available moisture for evaporation thus relatively low $r$ value as compared to the other $r$ values under = 1.2 m.s$^{-1}$ and 2.5 m.s$^{-1}$.

### 3.2 Evaporative demand ($E_d$) and discharge ($q_t$) relationship

The normality test (Table 3) was carried out on the observed discharge values for each applied evaporative demand scenario ($E_d$ = 0.16 mm.d$^{-1}$, 0.31 mm.d$^{-1}$ and 0.36 mm.d$^{-1}$) The respective sample data revealed high data skewness ($p<0.05^*$). The statistical analysis revealed that there were no significant differences among the means across the three $E_d$ categories

**Table 3. Summarised descriptive statistics for the induced $E_d$ and the observed $q_t$.**

| Evaporative Demand ($E_d$) (mm.d$^{-1}$) | 0.16 | 0.31 | 0.36 |
|---|---|---|---|
| Mean Discharge ($q_t$) (l.h$^{-1}$.m$^{-1}$) | 0.00139 | 0.0014 | 00.0015 |
| $p^*$ | $p<0.05$ | $p<0.05$ | $p<0.05$ |
| $p^{**}$ | $p>0.05$ | $p>0.05$ | $p>0.05$ |

$p^*$ represents the Shapiro $p$-value at 95% confidence interval (CI) and $p^{**}$ represents the non-parametric Kruskal-Wallis one-way ANOVA test.

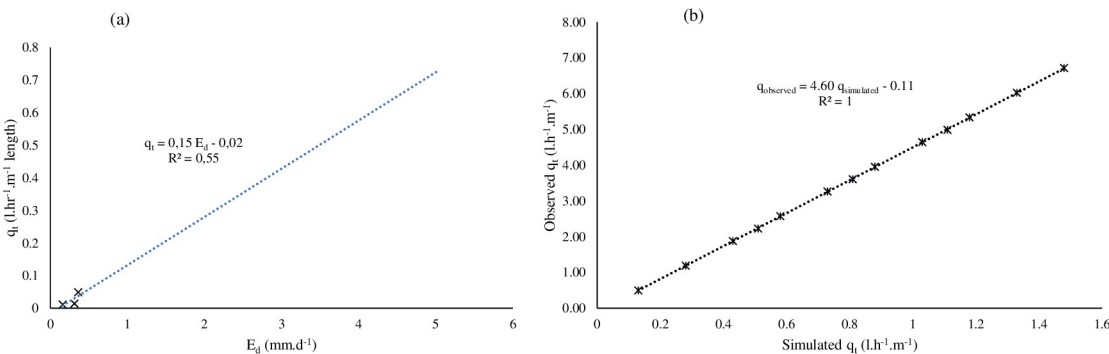

**Fig 2.** (a) Moistube™ discharge ($q_t$) at $0 \leq E_d \leq 7$ and (b) comparison between the simulated $q_t$ vs the observed $q_t$.

($p > 0.05^{**}$). Moistube™ discharge reaches a constant beyond a certain threshold time under the imposed negative pressure.

The effect of the negative pressure was determined by establishing a relationship between mass flow rates ($\dot{m}$) against the recorded $E_d$. The overall average $q_t$ vs $E_d$ was modelled as in Eq 8 ($R^2 = 0.55$) as shown in Fig 2(A) over selected time scales.

The equation represented average discharge ($q_t$) value for each respective $E_d$ sessions, i.e., the $E_d = 0.16$ mm.d$^{-1}$ had 1908 data points, whilst $E_d = 0.31$ mm.d$^{-1}$ had 1608 data points, and $E_d = 0.36$ mm.d$^{-1}$ had 120 data points. For comparative analysis purposes a normalized time scale ($0 \leq t^* \leq 1$) was used because the last $E_d = 0.36$ mm.d$^{-1}$ experimental conditions exceeded realistic temperature scenarios that a buried MTI lateral can experience.

$$q_t = 0.15E_d - 0.02 \qquad\qquad 8$$

where: $q_t$ = average discharge (l.h$^{-1}$.m$^{-1}$) across the respective $E_d$ sessions. A linear increase in $q_t$ relationship is observed as $E_d$ increases and under humid conditions where $E_d = 0$, MTI discharge $q_t = -0.02$ l.hr$^{-1}$.m$^{-1}$ length, which amounts to zero discharge. Abu-Zreig, Abe [23] in their study with ceramic pitcher pots recorded a linear discharge-evaporative demand relationship ($R^2 = 0.98$). MTI functionality closely resembles that of ceramic pitcher pots. Eq 12 characterizes laminar flow through a porous media (MTI) and it is a function of membrane surface area ($A$), flow path length ($L$), flow duration (irrigation interval) ($t$) [24, 25], and MTI hydraulic properties such as effective porosity of MTI lateral ($\varphi$), and inertial coefficient of MTI ($\delta$) [26]. The functional relationship is characterised by Eq 9. Eq 8 shows that at $E_d = 0$, MTI experiences zero discharge ($q_t = -0.02$), that is the MTI inertial coefficient facilitates an undisturbed fluid force.

$$q_{Ave} = f(A, \varphi, L, t, \delta) \qquad\qquad 9$$

The established relationship (Eq 12) is characterized by a one-sided $E_d$ sensitive limit as in Eq 10. The limits were informed by the FAO evaporating power scale [7]. The simulated $q_t$ data had a significant difference ($p < 0.05$) across the respective regions (humid, semi-arid and arid regions).

$$lim_{E_d \to 10} 0.15E_d - 0.02 = q_t \text{ for } 0 \leq E_d \leq 0.36 \qquad\qquad 10$$

The observed ($q_o$) (including the extrapolated values) and the simulated ($q_{sim}$) (see S1 Appendix) were plotted on a linear regression plot and yielded a correlation value of $R^2 = 1$ (Fig 2B). The model overestimated the $q_t$ ($PBIAS = 77\%$), thus, the limits for Eq 12 applicability were defined by Eq 10.

A higher $E_d$ resulted in high discharge rates. To assess the effects of $E_d$ on $q$ the study employed the relative discharge approach defined by Eq 11:

$$q_{rel} = \frac{(q_i - q_o)}{q_o} * 100 \qquad 11$$

where $q_{rel}$ = average relative discharge (%), $q_i$ = average discharge (l.h$^{-1}$.m$^{-1}$) at $t^* = 0 \le t^* \le 1$ h and $q_o$ = average initial discharge obtained at the beginning of the experiment., $t = 0$ (l.h$^{-1}$.m$^{-1}$).

For $E_d = 0.31$ mm.d$^{-1}$ there was a 10% decline in relative discharge ($q_{rel}$) whilst $E_d = 0.36$ mm.d$^{-1}$ recorded a 67% decline in $q_{rel}$ over a $t^* = 1$ h period. A high $E_d$ corresponded to a high $q_{rel}$. Yang, Tian [3] postulated that in the absence of a driving pressure MTI discharge is a function of matric potential or a negative pressure. Under $E_d = 0.16$ mm.d$^{-1}$, it was observed that over 10 h the discharge rose from 0.043 l.h$^{-1}$.m$^{-1}$ to 0.077 l.h$^{-1}$.m$^{-1}$. This effect can be attributed to a slow and gradual increase in VPD ($e_a - e$) within the PVC conduit that effected continuous and gradual discharges.

A near saturation scenario was observed from $t^* = 0.1$ hr to $t^* = 1$ hr under $E_d = 0.16$ mm.d$^{-1}$ which signified a protracted equilibrium scenario whereby $e_a \ge e$. For a buried Moistube™ lateral, continuous discharge is observed until water potential between soil-moisture and the water inside the MTI equilibrates. For applicability, to ensure continuous discharge beyond equilibrium points a net positive pressure is required to effect discharge as stated in studies [1, 27].

## 3.3. Discharge ($q$) vs time ($t$) relationship

The $q$ vs $t$ relationship was established on a normalised time-scale ($t^*$) (Fig 3). Statistical analysis revealed significant differences ($p < 0.05$) in $q$ over different $E_d$ scenarios. The normalised run-times for each experiment are shown in Table 4. The study established a functional relationship between $q$ and $t^*$ characterized by Eq 12.

$$q_t = K_t e^{-\beta t^*}, \text{ for } 0 < t < a_i \qquad 12$$

where: $q_t$ = time dependent discharge, $K_t$ = constant of proportionality dependent on MTI hydraulic properties, MTI surface area ($A$) flow path length ($L$), [28] and, $\beta$ = discharge

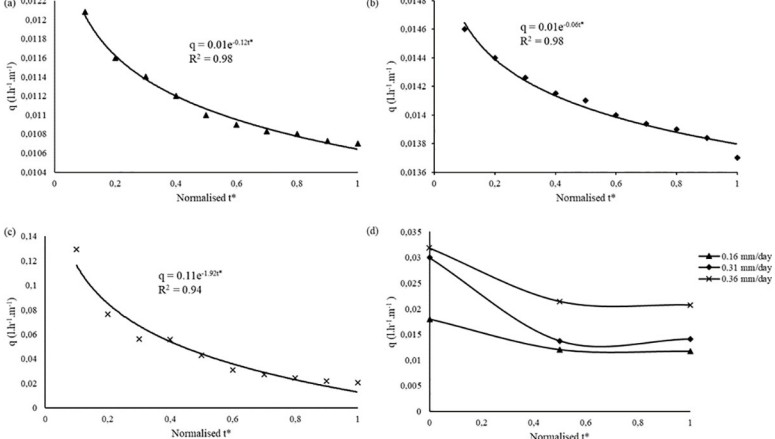

**Fig 3.** Discharge ($q_t$) vs. normalised time ($t^*$) relationship at (a) $E_d = 0.16$ mm.d$^{-1}$, (b) $E_d = 0.31$ mm.d$^{-1}$, (c) $E_d = 0.36$ mm.d$^{-1}$, and (d) combined $q$ vs $t^*$ plots.

**Table 4. Normalised time scale ($t^*$) vs $E_d$ for the $q_t$ vs $t^*$ plot.**

| | Normalised time scale ($t^*$) | | |
|---|---|---|---|
| | **0** | **0.5** | **1** |
| $E_d$ **session (mm.d$^{-1}$)** | **Actual run-times (h)** | | |
| 0.16 | 0.1 | 79.5 | 159 |
| 0.31 | 0.1 | 67 | 134 |
| 0.36 | 0.1 | 5 | 10 |

exponent, $t^*$ = normalised time for a specific induced $E_d$, and $q_i$ = normalised upper limit time range for each subsequent $E_d$, value. The relationship revealed that $q_t$ is time sensitive i.e. if MTI is continuously exposed to an imposed $E_d$, MTI discharges exhibits an exponentially decreasing trend to a point of stability at each respective normalised time.

For $E_d$ = 0.36 mm.d$^{-1}$ there was a steady decline in discharge (Fig 3C). The recorded flow rate variations were approximately 0.129 l.h$^{-1}$.m$^{-1}$ length from $t^*$ = 0.2 to $q_t$ = 0.021 l.h$^{-1}$.m$^{-1}$ length at $t^*$ = 1. Whereas for $E_d$ = 0.16 mm.day$^{-1}$ and $E_d$ = 0.31 mm.d$^{-1}$ discharge reached a stable steady state ranging from $q$ = 0.011 l.h$^{-1}$ length.m$^{-1}$ to 0.014 l.h$^{-1}$.m$^{-1}$ length at $t^*$ = 0.45 and $t^*$ = 0.8, respectively. These low discharge variations mimic a buried Moistube lateral experiencing minimal to zero discharge at low soil-water potential. Niu, Zhang [29] posited that at zero driving head a buried Moistube lateral reaches a stable steady state discharge after 48 hours. Also, The stable flow rates describe a near saturation phenomenon with in the PVC enclosure, thus the suction effect of the imposed negative pressure had little effect on the Moistube discharge since the observed discharges were lower than the nominal MTI discharge of 0.3 l.hr$^{-1}$.m$^{-1}$ length. Yang, Tian [3] asserted that discharge from a buried Moistube™ lateral reduces or stops when the matric potential of the surrounding soil approaches saturation i.e. when $\psi_{water} \leq \psi_{soil}$ less seepage from the Moistube tubing is anticipated and when $\psi_{water} = \psi_{soil}$ there is zero seepage. Once MTI is in equilibrium with its surrounding, discharge stops and only a positive driving pressure will induce discharge. A similar observation was made by Abu-Zreig, Abe [23].

For $E_d$ = 0.16 mm.d$^{-1}$ it was observed that there was uniform and steady decline in discharge rates from $t^*$ = 0.1 to $t^*$ = 0.5 (33% decrease in relative discharge rate), thereafter there was a steady state discharge rate ($t^*$ = 0.5 vs $t^*$ = 1) where the discharge varied from 0.011 l.h$^{-1}$.m$^{-1}$ length to 0.0107 l.h$^{-1}$.m$^{-1}$ length resulting in a $q_{rel}$ = 2.7%. The phenomenon can potentially be attributed to a stagnating VPD ($e_a - e$) over time. The steady state discharge rates were also observed on the $E_d$ = 0.31 mm.d$^{-1}$ after $t^*$ = 0.5. This phenomenon indicates a situation where VPD stabilizes such that evaporation occurs over an extensively wet MTI tubing i.e. wet environment evaporation ($E_w$) [8]. For a buried MTI tubing the slow stable irrigation water release would occur in a near saturation MTI–$\psi_{soil}$ continuum. From Fig 3, the $E_d$ = 0.36 mm.d$^{-1}$ discharge decreased rapidly due to the high drying power of the air which increased the VPD ($e_a - e$) resulting in a high $E_d$ that induces discharge. The findings concur with Abu-Zreig, Zraiqat [11] experiment with pitcher pots under constant head. The seepage rate of both buried pitcher pots and those in a controlled environment had a steady decrease as soil water increase. All three plots plateaued signifying a saturated micro-environment within the PVC conduit and consequently a decreased air suction effect.

## 4 Irrigation implication

When Moistube tubing is exposed to an evaporative demand a negative pressure develops that induces discharge. The phenomenon facilitates slow release of water thus allowing MTI tubing

to be utilized for optimal field water use efficiency (fWUE), i.e., slow releasing moisture as per crop water requirements. MTI is conceptually a continuous irrigation technique which potentially maintains the soil at saturation level thus, it is capable of maintaining soil moisture levels at field capacity (FC). This is instrumental in availing adequate crop water without having to fully saturate the soil [30]. This is reflected in the study wherein discharge plateaus (constant) under $E_d$ = 0.16 mm.d$^{-1}$ and $E_d$ = 0.31 mm.d$^{-1}$. The implication highlighted was that MTI can supply water without applying a driving pressure along an irrigation line. In addition, the negative pressure system can supply water in minute quantities just below crop water requirements.

The zero-pressure head discharge phenomenon saves on irrigation energy that irrigators otherwise would incur when using other irrigation systems. Climate change destabilizes planned crop water use patterns thus, MTI tubing can be adopted in arid and semi-arid regions for intermittent water application. The $\psi$ concept as investigated by the study in the form of an evaporative demand ($E_d$) can be utilized by MTI tubing to facilitate water saving by timeously controlling discharge rates and thus availing water to crops when the soil-moisture drops beyond a critical level. This helps lower the yield penalties on crops that are sensitive to water deficit.

## 5 Recommendations

The study was an open-air experiment; hence it is recommended that the study be carried on an actual buried MTI tubing wherein the soil matric potential ($\psi$) are present and influence soil-moisture movement. The $E_d$ vs $q_t$ relationship should be experimentally explored to define the actual limits of the MTI operation and $q$ variations for high evaporating ($ET_o$) power of the atmosphere.

## 6 Conclusions

The study mimicked a buried MTI tubing, wherein the VPD represented the soil matric potential. At various evaporative demand scenarios, the MTI tubing released moisture at diminishing rates, thus rejecting the null hypothesis. An increase in $E_d$ subsequently resulted in an increased MTI discharge, a characteristic likely to be observed under arid climatic conditions. The steady plateauing discharge curve under low $E_d$ (0.16 mm.d$^{-1}$) represented a humid environment where evaporation was occurring on a relatively wet surface. The functional $E_d$−$q_t$ relationship revealed that at $E_d$ = 0 there is no MTI discharge scenario, therefore, in humid regions there is need to incorporate pumping units to drive irrigation water and effect MTI discharge. Likewise, in arid regions where $E_d$ > 7 mm.d$^{-1}$ irrigators can capitalise on the MTI negative pressure discharge capability thus minimising energy costs. The approach can be used to model irrigation schedules based on soil matric potential ($\psi$), which subsequently avails irrigation water as per crop water requirements (CWRs); thus, improving water use efficiency (WUE). MTI tubing performs according to the conceptual design, where it releases moisture at zero positive pressure head.

## Supporting information

**S1 Appendix. Data for the computed $q_{Ave}$ values using Eq 10.**
(DOCX)

## Author Contributions

**Conceptualization:** Tinashe Lindel Dirwai, Aidan Senzanje, Tafadzwanashe Mabhaudhi.

**Data curation:** Tinashe Lindel Dirwai.

**Formal analysis:** Tinashe Lindel Dirwai, Aidan Senzanje, Tafadzwanashe Mabhaudhi.

**Funding acquisition:** Aidan Senzanje.

**Investigation:** Tinashe Lindel Dirwai.

**Methodology:** Tinashe Lindel Dirwai, Aidan Senzanje, Tafadzwanashe Mabhaudhi.

**Supervision:** Aidan Senzanje, Tafadzwanashe Mabhaudhi.

**Validation:** Tinashe Lindel Dirwai.

**Visualization:** Aidan Senzanje, Tafadzwanashe Mabhaudhi.

**Writing – original draft:** Tinashe Lindel Dirwai.

**Writing – review & editing:** Tinashe Lindel Dirwai, Aidan Senzanje, Tafadzwanashe Mabhaudhi.

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
