## [Decision Letter · Decision Letter 0]

2 Aug 2020

PONE-D-20-20136

Moistube ™ Irrigation (MTI) Discharge Under Variable Evaporative Demand

PLOS ONE

Dear Dr. Dirwai,

Thank you for submitting your manuscript to PLOS ONE. After careful consideration, we feel that it has merit but does not fully meet PLOS ONE’s publication criteria as it currently stands. Therefore, we invite you to submit a revised version of the manuscript that addresses the points raised during the review process.

We look forward to receiving your revised manuscript.

Kind regards,

Vassilis G. Aschonitis

Academic Editor

PLOS ONE

Journal Requirements:

2.Thank you for stating the following in the Financial Disclosure section:

[The author(s) received no specific funding for this work.].   

We note that one or more of the authors are employed by a commercial company: VarMac Consulting Engineers

3.Thank you for stating the following in the Funding Section of your manuscript:

[This work was supported by the Agricultural Research Council (ARC) – Drainage Project and the University of KwaZulu-Natal – Pietermaritzburg, South Africa.]

 [The author(s) received no specific funding for this work.]

Reviewers' comments:

Reviewer's Responses to Questions

**Comments to the Author**

1. Is the manuscript technically sound, and do the data support the conclusions?

Reviewer #1: Partly

Reviewer #2: Yes

2. Has the statistical analysis been performed appropriately and rigorously? 

Reviewer #1: No

Reviewer #2: N/A

3. Have the authors made all data underlying the findings in their manuscript fully available?

Reviewer #1: No

Reviewer #2: No

4. Is the manuscript presented in an intelligible fashion and written in standard English?

Reviewer #1: No

Reviewer #2: Yes

5. Review Comments to the Author

Reviewer #1: I have read this article titled ‘Moistube™ Irrigation (MTI) Discharge Under Variable Evaporative Demand’ carefully. It’s interesting to research theories of negative pressure irrigation in an artificial evaporative way and the methods it used could be a reference to similar studies in future . However, there existed many problems needing revised completely before taking into consideration for acceptance.

1. Experiments, statistics, and other analyses are not performed to a high technical standard and are not described in sufficient detail as well. For instance, the details of experiment equipments were not clearly described, which was crucial to help readers to understand the paper. Statistics was not performed rigorously and No differences comparisons between treatments.

2. The article is not written in standard English, and Language is redundance, not concise.

3. The other suggestions had been labeled on the paper, please check it one by one.

Reviewer #2: The issue of Irrigation Discharge Under Variable Evaporative Demand is an important area of research that is not resolved in the literature. Therefore, the topic of this manuscript is appropriate for the journal and of interest to the community. I did find that the organisation of the manuscript, methods as described, and presentation of the results makes it difficult to assess the novelty of the manuscript’s contribution and replicate the experiments.  I suggest that this paper be resubmitted subject to moderate revision before consideration for publication.

Comments are presented below.

1. First of all, the hypothesis is not well defined. The authors are advised to highlight the hypothesis of their work in the broader context so that it can address the general audience. 

2. The introduction is incomplete and requires a significant overhauling. First, eliminate the equations and address the shortfalls of the previous research works verbally. The equations make it hard to understand how the novel work covers the shortfalls of the previous works.

3. In defining the study area, instead of information about the Laboratory's location, the information on variables used, their characteristics and the variability of these characteristics should be presented. The degree to which these characteristics vary across the study area would have an effect on the results so this should be reported.

4.  In the conclusion and recommendation section, the authors have summarised their conclusions in the first paragraph. Instead of deliberating each parameter, they focused on the acceptance/ rejection of the null hypothesis. I recommend the authors to briefly deliberate each selected parameter (or what they call "various evaporative demand scenarios") before coming to the null hypothesis. 

I wish authors all the best.

6. PLOS authors have the option to publish the peer review history of their article (what does this mean?). If published, this will include your full peer review and any attached files.

Reviewer #1: No

Reviewer #2: No

---

## [Author Response · Author response to Decision Letter 0]

25 Sep 2020

The authors have attached a Review-author log. The log attends to each and every comment/query raised by the reviewer.

---

## [Decision Letter · Decision Letter 1]

28 Oct 2020

PONE-D-20-20136R1

Moistube ™ Irrigation (MTI) Discharge Under Variable Evaporative Demand

PLOS ONE

Dear Dr. Dirwai,

Thank you for submitting your manuscript to PLOS ONE. After careful consideration, we feel that it has merit but does not fully meet PLOS ONE’s publication criteria as it currently stands. Therefore, we invite you to submit a revised version of the manuscript that addresses the points raised during the review process.

We look forward to receiving your revised manuscript.

Kind regards,

Vassilis G. Aschonitis

Academic Editor

PLOS ONE

Reviewers' comments:

Reviewer's Responses to Questions

**Comments to the Author**

1. If the authors have adequately addressed your comments raised in a previous round of review and you feel that this manuscript is now acceptable for publication, you may indicate that here to bypass the “Comments to the Author” section, enter your conflict of interest statement in the “Confidential to Editor” section, and submit your "Accept" recommendation.

Reviewer #3: All comments have been addressed

Reviewer #4: (No Response)

2. Is the manuscript technically sound, and do the data support the conclusions?

Reviewer #3: Yes

Reviewer #4: Partly

3. Has the statistical analysis been performed appropriately and rigorously? 

Reviewer #3: Yes

Reviewer #4: No

4. Have the authors made all data underlying the findings in their manuscript fully available?

Reviewer #3: No

Reviewer #4: Yes

5. Is the manuscript presented in an intelligible fashion and written in standard English?

Reviewer #3: Yes

Reviewer #4: No

6. Review Comments to the Author

Reviewer #3: I have no comments to make. I think that the authors have answered in an acceptable way all the questions that have been caused by their article.

Reviewer #4: The Moistube irrigation (MTI) system is a very interesting and promising way for the increment of water use efficiency in agriculture, therefore I have crucial concerns about the structure of the manuscript, the description of the experiment and the interpretation of the results. I state some specific objections and some general as well.

In the revised manuscript the first equation appears is Equation 5, nevertheless I also read carefully the first submission in order to have a clear aspect.

122: dot is missing

127: dot is missing

196: “to be normalised ,tmax” make it to be “to be normalised, tmax”

Equation (6): Are not all variables explained (Lu)

222: “The results (Table 2) revealed a positive correlation amongst the variables ua, r, VPD and Ed.” I see negative correlation between r and Ed.

234-235: This is not understandable.

Table 3: “Ed” make it to be “Ed”

Table 3: “00.0015”

242-245: “The MTI discharge under variable Ed was characterized by a power function (R2 = 0.62) as shown in Figure 2 over selected time scales. Fig 2 (a) Moistube™ discharge (qt) at 0 ≤ Ed ≤ 7 and (b) comparison between the simulated qt vs the observed qt.” What is this power function? Maybe you refer to Equation 16, so it is confusing for reader while at this point hasn’t seen the equation. I don’t see R2=0.62 in Fig. 2. I also don’t see power function but a linear relation between qt and Ed.

248-251: I am pretty confused of the procedure you followed. As far as I understand (I’m not sure about that), you calculated the qt with equation: qt=0.15Ed-0.02 according to the regression it seems in Fig. 2. This equation was built by using the average values of measured qt for the respective daily Ed values (i.e. 3)? Do you think the number of points you used for the regression modeling are adequate?

292-294: “These observed variations in between = 0.31 mm.d-1 and = 0.36 mm.d-1 are attributed to high values due to increased drying power of the air in the conduit, hence effecting high discharges.” This is confusing.

In Figure 3 points of different kinds should be corrected.

321-325: It is not clarified well the graph you refer to (a, b, c or d). There are divergencies between the values that are referred and Figure 3.

Concerning the “Results and Discussion” section, there are ambiguities which confuse the reader. I consider that the experiment was extensive, but I actually didn’t understand very well the interpretation of the results.

365-366: How the applicability and performance of the particular experiment is connected to the maintenance of soil water at 80-90% of the field capacity levels? At this point field capacity is referred for the first time.

7. PLOS authors have the option to publish the peer review history of their article (what does this mean?). If published, this will include your full peer review and any attached files.

Reviewer #3: No

Reviewer #4: No

---

## [Author Response · Author response to Decision Letter 1]

11 Nov 2020

The authors have responded to the reviewer comments. We have attached a reviewer-author comments log that addresses each individual query raised by the reviewers.

---

## [Decision Letter · Decision Letter 2]

30 Nov 2020

Moistube ™ Irrigation (MTI) Discharge Under Variable Evaporative Demand

PONE-D-20-20136R2

Dear Dr. Dirwai,

We’re pleased to inform you that your manuscript has been judged scientifically suitable for publication and will be formally accepted for publication once it meets all outstanding technical requirements.

Kind regards,

Vassilis G. Aschonitis

Academic Editor

PLOS ONE

Additional Editor Comments (optional):

Reviewers' comments:

Reviewer's Responses to Questions

**Comments to the Author**

1. If the authors have adequately addressed your comments raised in a previous round of review and you feel that this manuscript is now acceptable for publication, you may indicate that here to bypass the “Comments to the Author” section, enter your conflict of interest statement in the “Confidential to Editor” section, and submit your "Accept" recommendation.

Reviewer #3: All comments have been addressed

Reviewer #4: All comments have been addressed

2. Is the manuscript technically sound, and do the data support the conclusions?

Reviewer #3: Yes

Reviewer #4: Yes

3. Has the statistical analysis been performed appropriately and rigorously? 

Reviewer #3: Yes

Reviewer #4: Yes

4. Have the authors made all data underlying the findings in their manuscript fully available?

Reviewer #3: Yes

Reviewer #4: Yes

5. Is the manuscript presented in an intelligible fashion and written in standard English?

Reviewer #3: Yes

Reviewer #4: Yes

6. Review Comments to the Author

Reviewer #3: (No Response)

Reviewer #4: (No Response)

7. PLOS authors have the option to publish the peer review history of their article (what does this mean?). If published, this will include your full peer review and any attached files.

Reviewer #3: No

Reviewer #4: No

---

## [Editor Report · Acceptance letter]

4 Dec 2020

PONE-D-20-20136R2 

Moistube Irrigation (MTI) Discharge Under Variable Evaporative Demand 

Dear Dr. Dirwai:

I'm pleased to inform you that your manuscript has been deemed suitable for publication in PLOS ONE. Congratulations! Your manuscript is now with our production department. 

Kind regards, 

on behalf of

Dr. Vassilis G. Aschonitis 

Academic Editor

PLOS ONE